# Quinazoline Based HDAC Dual Inhibitors as Potential Anti-Cancer Agents

**DOI:** 10.3390/molecules27072294

**Published:** 2022-03-31

**Authors:** Jyothi Dhuguru, Ola A. Ghoneim

**Affiliations:** 1Mitchell Cancer Institute, University of South Alabama, 1660 SpringHill Ave., Mobile, AL 36604, USA; 2College of Pharmacy and Health Sciences, Western New England University, 1215 Wilbraham Road, Springfield, MA 01119, USA; ola.ghoneim@wne.edu

**Keywords:** quinazoline, histone deacetylases (HDAC), cancer, kinase receptor, multi drug resistance, kinase inhibitor, phosphoinosityl kinase receptor (PI3K), bromo domain and extra-terminal domain (BET), epidermal growth factor receptor (EGFR), G9a/GLP receptor, vascular endothelial growth factor receptor (VEGFR)

## Abstract

Cancer is the most devastating disease and second leading cause of death around the world. Despite scientific advancements in the diagnosis and treatment of cancer which can include targeted therapy, chemotherapy, endocrine therapy, immunotherapy, radiotherapy and surgery in some cases, cancer cells appear to outsmart and evade almost any method of treatment by developing drug resistance. Quinazolines are the most versatile, ubiquitous and privileged nitrogen bearing heterocyclic compounds with a wide array of biological and pharmacological applications. Most of the anti-cancer agents featuring quinazoline pharmacophore have shown promising therapeutic activity. Therefore, extensive research is underway to explore the potential of these privileged scaffolds. In this context, a molecular hybridization approach to develop hybrid drugs has become a popular tool in the field of drug discovery, especially after witnessing the successful results during the past decade. Histone deacetylases (HDACs) have emerged as an important anti-cancer target in the recent years given its role in cellular growth, gene regulation, and metabolism. Dual inhibitors, especially based on HDAC in particular, have become the center stage of current cancer drug development. Given the growing significance of dual HDAC inhibitors, in this review, we intend to compile the development of quinazoline based HDAC dual inhibitors as anti-cancer agents.

## 1. Introduction

Cancer is the most fatal disease affecting millions of people worldwide and has resulted in 9,958,133 mortalities in 2020 alone [1]. Among the various types of cancers that include breast, lungs, liver, prostrate, esophagus, cervix, etc., the most common and frequently encountered are the breast, prostate, and lung cancers [1,2]. There are several types of treatments for cancer such as chemotherapy, hormone therapy, radiation therapy, photodynamic therapy, immunotherapy, hyperthermia, stem cell therapy, targeted therapy, and, in some cases, surgery is the best option [3]. Despite the numerous drug classes available in the market for cancer treatment, challenges like multidrug resistance by cancer cells, as well as the intolerable side effects of drugs, hindered the efficiency of chemotherapy, and add an overwhelming and frustrating experience for the patients undergoing chemotherapy.

## 2. History and Significance of Quinazolines

Heterocyclic compounds and, in particular, nitrogen bearing compounds have tremendous values in the field of medicinal chemistry. With a plethora of potential pharmacological activities, nitrogen containing derivatives continue to enthrall scientists worldwide with their therapeutic properties, and this is the reason why heterocyclic compounds are the building blocks of many drugs and pharmaceutical products [4,5,6,7]. Among the nitrogen containing heterocyclic aromatic scaffolds, quinazolines are considered as privileged compounds. Quinazoline scaffolds represent the ubiquitous class of heterocyclic compounds that are found in over 200 naturally occurring alkaloids [4,8,9]. They have gained wide recognition and also constitute the most studied moieties in heterocyclic chemistry. Ever since their discovery, these fused ring compounds have been the center of interest for medicinal chemists due to their attractive biological and pharmacological applications. Although quinazolines were first recognized for their anti-malarial activity, their derivatives later found diverse applications such as antimicrobial, anti-fungal, anti-convulsant, anti-tumor, anti-hypertensive, anti-diabetic, anti-inflammatory, anti-HIV, and kinase inhibition [10,11,12,13]. Besides their pharmacological applications, structural modification of these compounds rendered them attractive fluorescent probes, which was vastly exploited by researchers expanding their applications in to optical imaging and sensing [14,15].

## 3. Structure, Isomerism, and Synthesis of Quinazolines

Quinazoline is a bicyclic heterocyclic compound formed from the fusion of two six-membered aromatic rings of benzene and pyrimidine. These compounds belong to 1,3-benzodiazine (diazonaphthalene) category which contains two nitrogen atoms. Based on the position of nitrogen atoms, the quinazoline ring can exist in four isomeric forms such as quinazoline, quinoxaline, cinnolines, and pthalazines (Figure 1). The oxidized form of quinazolines are called quinazolinones, which can be further classified into 2(1H)-quinazolinones, 4(3H)-quinazolinones, and 2,4(1H,3H)-quinazolinedione based on the position of the keto group (Figure 1) [16].

The first synthesis of quinazoline was accomplished by Bischler and Lang in the year 1895 [17]. Although these fused ring compounds were known as ‘benzo-1,3-diazine’ in the beginning of their discovery, Weddige later named them as ‘quinazolines’ in the year 1887. The discovery of febrifugine, an anti-malarial drug marked the beginning of an extensive research on the pharmacological activities of quinazolines [18]. The history of quinazoline dates back to 1869, when Griess synthesized 2-cyano-3,4-dihydro-4-oxoquinazoline, the first derivative of quinazoline. Vasicine (peganine), known for its bronchodilatory and anti-tuberculosis activity, was the first quinazoline alkaloid isolated from *Justicia adhatoda* [19].

Owing to its ubiquitous nature, synthetic versatility and widespread biological and pharmacological applications, numerous synthetic methods have come to light for the facile synthesis of quinazoline and its derivatives both for therapeutic and imaging purposes [15]. The most notable among them is the copper-catalyzed intramolecular N-arylation, Rhodium-catalyzed ortho-amidation, cyclocondensation, tetrabutyl ammonium fluoride promoted cyclization of N-imidoyl-o-alkynylanilines, Niementowski reaction, and benzotrizol-1-yloxytris(dimethylamino) phosphonium hexafluorophosphate (BOP)-mediated ring closures [8,17,20].

## 4. Quinazolines as Anti-Cancer Agents

Quinazolines are widely acclaimed for their efficiency as anti-cancer agents. Several quinazoline based kinase inhibitors such as afatinib, lapatinib, gefitinib, dacomitinib, tucatinib, and erlotinib are FDA approved for cancer treatment which prove their therapeutic efficiency as anti-tumor agents (Figure 2). Various approaches were proposed to explain the mechanism of their anti-tumor activity, which can be attributed to one or more of the following pathways: (1) Tubulin inhibition, (2) Epidermal growth factor receptor (EGFR) inhibition, (3) Inhibition of DNA damage/repair mechanism, and (4) Inhibition of thymidylate synthetase [21,22].

## 5. Histone Deacetylase Inhibition as Anti-Cancer Agents

Histone deacetylases (HDACs) were discovered by Vincent Allfrey who identified that this set of enzymes can remove acetyl groups from histones [23]. Based on the homology sequence and subcellular localization, HDACs are divided in to four classes—Class I (HDAC-1, 2, 3 and 8), Class II (HDAC-4, 5, 6, 7, 9 and 10), Class III (SERT protein 1–7), and Class IV (HDAC-11). All the HDACs are Zn^+2^ dependent enzymes except Class III that are NAD^+^ dependent and not Zn^+2^ dependent [24]. The fact that these enzymes are overexpressed in several human cancers and are involved in several critical biological events such as DNA damage repair, regulation of transcription, cell cycle, autophagy, and other stress responses besides deacetylation of histone and non-histone residues rendered these substrates as attractive cancer-therapeutic targets [25,26,27]. Therefore, the development of small molecule HDAC inhibitors has gained a lot of attention in the past decade [28,29,30,31]. Currently, there are five FDA approved HDAC inhibitors available for cancer treatment: Panobinostat, vorinostat (SAHA), belinostat (PXD-101), romidepsin (FK-228), and tucidinostat (HBI-8000) (Figure 3).

## 6. Multi-Targeted Therapy and Multi-Drug Resistance

As commonly known, drug resistance is the one of the major challenges encountered in the cancer treatment by chemotherapy [3,32]. Currently, anti-cancer drugs available in the market suffer from two major challenges: toxicity and drug resistance [33]. Based on the time of development, resistance to cancer drugs can be either intrinsic or acquired and can be the outcome of several factors. Some of them include drug metabolism, sequestration of anti-cancer drugs in lysosomes, overexpression of ATP binding cassette (ABC) transporters leading to the extrusion of anti-cancer drugs, alterations in the tumor micro-environment, genetic mutations, drug inactivation, and inhibition of apoptosis just to name a few [34,35,36]. Given the complexity and heterogeneity in the tumor invasion and progression and considering the quick adaptation of cancer drugs to any single chemotherapeutic agent, cocktail or multi-drug therapy which includes treatment with a combination of two or more drugs has become a most common approach in the recent years to treat tumor malignancies, in order to circumvent the drug resistance and improve the safety and efficacy of cancer treatment with less toxicity, multi-targeted therapy has evolved as an effective tool in the field of drug discovery [3,37,38].

However, multi-drug resistance (MDR) is another major hurdle encountered in cancer chemotherapy, which is a phenomenon in which cancer cells develop resistance to structurally or pharmacologically unrelated drugs. Several strategies were developed to overcome the MDR that include inhibition of membrane transport proteins (ABC), gene silencing by turning off the drivers of MDR, transcriptional regulation, targeted delivery of anti-cancer drugs, and hybrid/chimeric drug approaches are some of them [3,32].

## 7. Synergistic Effect of the Dual Inhibition/Hybrid Approach

Hybrid drugs are obtained by the fusion of one or more pharmacophore units or bioactive molecules using a molecular hybridization technique [37]. The resulting hybrid or chimeric molecules have a chemical structure different from the individual parent molecules and are connected by a labile spacer or linker. Several reports appeared in the literature which indicate that the hybrid derivatives have improved affinity, efficacy, and less toxicity compared to the parent entities. Hybrid drugs can be an alternative path to avoid MDR.

The pharmacophoric scaffold of HDAC inhibitor usually consists of a capping group, a zinc-binding group—which is most commonly hydroxamate/hydroxamic acid—and a proper linker (Figure 4). The hydroxamate group is critical for efficient binding of the small molecule inhibitor to the enzyme. However, given the multi-drug resistance and the dearth of potent HDAC inhibitor with high efficacy, development of dual HDAC inhibitors has evolved as a lucrative approach to cancer treatment [39,40,41]. Several reports came to light in the past decade highlighting the pharmacological significance of the fusion of the quinazoline pharmacophore with various FDA approved HDAC inhibitors to harness the synergistic effects of these two pharmacophoric units.

The main focus of this review is to shed light on the quinazoline based dual HDAC inhibitors with other cancer epigenetic targets. Given the pharmacological significance of quinazoline pharmacophore and the growing importance of HDAC inhibitors in cancer treatment, we herein compiled the drug design and activity of quinazoline based dual inhibitors targeting HDAC enzyme with itself and other therapeutic targets such as PI3K, EGFR, VEGFR, and G9a/GLPand will be fully discussed in the following sections.

### 7.1. Quinazoline Based Dual Inhibition of HDAC1 and HDAC6 Enzymes

In order to develop potent dual inhibitors of HDAC1 and 6, Chen et al. designed and synthesized nearly fifty-eight quinazoline based derivatives [42]. Of all the derivatives, two compounds (**1** and **2**) in particular demonstrated synergistic inhibition of HDAC1 and 6 (Figure 5) [42]. Both the compounds exhibited 10-fold selectivity to other HDACs in addition to tubulin acetylation and induction of histone H3 acetylation in Hela cells. Compound **1** exhibited IC_50_ values of 31 nM and 16 nM against HDAC1 and 6, respectively, whereas compound **2** showed IC_50_ of 37 nM and 25 nM against HDAC1 and 6, respectively (Table 1).

Remarkably, both the compounds demonstrated strong anti-proliferative activity against various tumor cell lines including solid tumors (HepG2, M-M-231, MCF-7, H1975, H460, and Hela cells) with IC_50_ less than 40 nM values and importantly in hematological tumors’ cells (U266 and RPMI8226) with IC_50_ > 1 Nm (Table 2) [42].

Compound **1** in particular showed significant inhibition of tumor growth in a resistant MCF-7/ADR xenograft animal model with no significant changes in the body weight and behavior [42].

### 7.2. Quinazoline Based Dual Inhibition of HDAC and PI3K

Phosphoinositide 3-kinases (PI3Ks) are a class of lipid kinases that are involved in the regulation of a range of cellular events such as cell division, apoptosis, cell growth, DNA repair, angiogenesis, cell survival, and motility. These are key signaling intermediates in the PI3K/AKT/mTOR signaling pathways [43]. Dysregulation of PI3K pathway is observed in several different types of cancer making them an attractive target for the development of cancer therapeutics [44,45,46]. PI3K inhibitors find applications in the treatment of variety of human cancers such as non-small cell lung cancer (NSCLC), breast cancer, head and neck squamous cell carcinomas (HNSCC), and other solid tumors [43,47]. Duvelisib, pictilisib, taselisib, alpelisib, idelalisib, buparlisib, and copanlisib are some of the PI3K selective inhibitors in various stages of clinical development for the treatment of different types of cancers (Figure 6) [48,49].

Recently, Zhang et al. designed a series of dual PI3K and HDAC inhibitors incorporating a quinazoline based PI3K pharmacophore (compound **3**) with the hydroxamic acid targeting the HDAC (Figure 7) [44,50].

The resulting hybrid scaffold can be effectively tuned to modulate the activities of PI3K and HDAC by varying the substitutions at the 3-position of methoxy pyridine of the PI3K inhibitor and the linker length of hydroxamate, respectively, as shown in Figure 7 [50].

When the resulting hybrid derivatives were tested for potency in various cancer cell lines, they found that two compounds (**4** and **5**)-compound **4** with an alkyl linker and compound **5** with a pyrimidine linker showed excellent HDAC and PI3K inhibition compared to the reference HDAC inhibitor SAHA and PI3K inhibitor: BKM120, respectively (Figure 8). In addition to the HDAC and PI3K inhibitor, activity was also compared to a dual HDAC/PI3K inhibitor: CUDC-907 as shown in Table 3.

Compound **5** in particular demonstrated excellent anti-proliferative activity against several cancer cell lines with IC_50_ values ranging from 0.05–2.7 µM range (Table 4) [50]. Both the compounds showed target modulation such as increased histone-H3 acylation and decreased p-AKT in addition to fast clearance and poor bioavailability similar to other HDAC inhibitors. On further investigation of the mechanism of action in hematologic tumors, it was revealed that compound **4** caused induction of apoptosis and G1-phase arrest in vitro and in vivo by the regulation of PI3K and HDAC signaling pathways [51].

Using a similar approach, Thakur et al. developed a series of dual PI3K and HDAC inhibitors by integration of HDAC targeting hydroxamic acid to a known PI3K inhibitor drug (idelalisib) through an optimized linker (Figure 9) [52]. The resulting dual inhibitors were identified to be potent against multiple cancer lines with excellent selectivity against PI3K γ, δ isoforms, and HDAC6 enzymes (Table 5).

SAR studies conducted against a panel of 60 cancer cell lines clearly demonstrated the selectivity and potency of most of the derivatives against PI3Kγ, δ, and HDAC6 enzymes.

Though most of the inhibitors exhibited nanomolar potency against the target enzymes (IC_50_ < 10 nM), one compound, notably compound **6**, was found to be the best of all the derivatives, as it showed remarkable anti-proliferative activity and high potency against a variety of cancer cell lines including NSCLC, leukemia, breast cancer, renal cancer, CNS cancer, and melanoma (Table 6, Figure 10).

The potency of compound **6** was further demonstrated from the induction of cell death via necrosis in FLT3-ITD mutant and FLT3-inhibitor resistant cell lines and primary blasts from acute myeloid leukemia (AML) patients. In addition to excellent pharmacokinetic profile and the dual inhibition, compound **6** was further evaluated using cellular thermal shift assay (CETSA) in MV411 cell line.52

### 7.3. Quinazoline Based Dual Inhibition of HDAC and BET

The bromodomain and extra-terminal domain (BET) family consists of four types of proteins: bromodomain containing protein 2 (BRD2), BRD3, and BRD4 that are found to be ubiquitous while bromodomain testis specific protein (BRDT) is localized to testes and oocytes [53]. BET family proteins are an important set of proteins involved in the acetylation of histone proteins and key players in cellular events such as cell-division, cell growth, apoptosis, and in the regulation of gene expression [54,55]. BET proteins are critical in the transcription of oncogenes and their dysregulation was implicated in several diseases including cancer as a consequence of triggering of oncogene production [56,57]. Consequently, the development of inhibitors targeting BET family members has evolved as a promising approach to treat specific cancers [58,59,60,61] Several small molecule BET inhibitors were developed in the recent years with some of them even entering clinical trials for cancer therapies such as I-BET762/GSK-525762A, TEN-010, JQ1, ODM-207, ABBV-075, and a quinazolone based BET inhibitor—Apabetalone (RVX-208) developed by Resverlogix (Figure 11) [62,63].

Reports have indicated that BET and HDAC inhibitors induce similar genes and synergize to kill Myc-linked murine lymphoma [55]. In this direction, several reports came to light to harness the collaborative benefits of HDAC/BET proteins [64]. Fiskus and team found that co-treatment of BRD4 antagonist (JQ1) and HDAC inhibitor panobinostat showed promising results against human acute myeloid leukemia (AML) and the co-treatment was more effective in the induction of apoptosis than individual treatment with either of the inhibitors [65].

Based on all of the above findings, Shao and his team developed a series of dual HDAC/BRD4 inhibitors by amalgamating the structural features of RVX-208, RVX-OH, and SAHA (Figure 12) [66]. Their selection was based on the fact that RVX-208 exhibits preferential binding (over 25-fold) towards BD2 (second bromo domain of BET protein) than BD1. On the other hand, RVX-OH, although lacking the selectivity towards the BET proteins comes with an advantage to form a hydrogen bond with N140 due to the free hydroxy group of its phenyl ring that acts as an acetyl-lysine mimetic group. In order to harness the structural advantages offered by both the pharmacophores of HDAC and BRD4 inhibitors, researchers used various types of linkers to connect the zinc-binding hydroxamate group with the free hydroxy group of the phenyl ring [66].

During their studies, research team found that RVX-208 had no biological activity on HDAC1 and similarly vorinostat (SAHA) by itself showed no activity against BRET proteins (BRD4/BD2), whereas the amalgamate derivatives (**7** and **8**, Figure 13) showed potent activity against HDAC1 and BRD4 with IC_50_ values of 32 and 204 nM, respectively, against HDAC and IC_50_ of 225 and 401 nM against BRD4, respectively (Table 7).

Most remarkably, compound **8** exhibited more pronounced growth inhibition of AML cells compared to single inhibitors of RVX-208 or SAHA alone, with an IC_50_ of 0.38 µM demonstrating the potency of dual inhibitors (Table 8) [66].

### 7.4. Quinazoline Based Dual Inhibition of HDAC and EGFR

Epidermal growth factor receptor (HER1/ErbB1) belongs to the ErbB receptor tyrosine kinase family [14,67]. Extracellular ligand binding results in the activation of the receptor leading to homodimerization or heterodimerization of the ErbB receptor, which ultimately activates multiple signal transduction pathways such as the mitogen activated protein kinase (MAPK) and PI3K pathways.

Dysregulation of EGFR is encountered in several types of cancers including head and neck, breast, lung cancers, prostrate, ovarian, glioblastomas, and other solid tumor malignancies [14]. Overexpression or dysregulation can be the outcome of relocation, mutations (deletions) in the EGFR gene, or aberrant signaling [68,69,70]. Consequently, small molecule inhibitors targeting EGFR have evolved as an attractive strategy to control the tumor growth and treatment of some types of cancers [22,67,71]. Extensive research is focused on the development of small molecule inhibitors and some have already been approved by FDA for the cancer treatment such as bosutinib, dasatinib, erlotinib, gefitinib, lapatinib, sunitinib, pazopanib, vemurafenib, and crizotinib (Figure 2 and Figure 14) [72]. Among these drugs, erlotinib, gefitinib, and lapatinib are the quinazoline based kinase inhibitors (Figure 2) [15,73].

Given the importance of HDAC inhibitors in arresting the cell growth, differentiation and apoptosis, the concept of dual inhibition of HDAC and EGFR is receiving attention from drug discovery scientists worldwide. Studies have demonstrated the successful design of dual inhibitors incorporating the pharmacophores of HDAC and EGFR inhibitors and obtained hybrid derivatives with strong anti-proliferative activity.

In this direction, Mahboobi et al. developed dual inhibitors of HDAC and EGFR by integrating the pharmacological activity of HDAC (Class I/II) enzyme inhibition with EGFR/HER2 kinase inhibition (Figure 15) [74].

For the drug design, their team selected hydroxamic acid and benzamide motifs as the zinc binding hydrophobic region of the HDAC inhibitors and integrated it with the 4-amino-substituted quinazoline pharmacophore to design the desired chimeric scaffolds.

As lapatinib was an established EGFR inhibitor, it was selected as a scaffold to design the dual inhibitors [75]. The resulting derivatives were tested for the activity of HDAC and protein kinase inhibition in various biochemical assays and EGFR or HER2 overexpressing cancer cell lines were further used to determine the cytotoxicity of the chimeric derivatives (Table 9 and Table 10).

Two hybrid derivatives, namely compounds **9** and **10**, were identified to be best in terms of selectivity and potency of HDAC and EGFR/HER2 inhibition (Figure 16). When the activity was tested in the head and neck cancer cell lines CAL 27, compound **10** resulted in concurrent inhibition of EGFR and induction of histone H3 acetylation (Table 11).

Detailed cytotoxicity studies revealed that the chimeric inhibitors with benzamide motif were not as potent as those integrated with hydroxamic acid, as the former ones failed to induce the HDAC inhibition although they demonstrated kinase inhibition. Compound **10** with hydroxamic acid moiety demonstrated a dual biological activity with an IC_50_ of 0.59 μM (Table 11) [74].

The same group developed another set of dual inhibitors targeting HDAC class I and II enzymes and EGFR/HER2 kinases (Figure 17) [76]. Although the pharmacological targets remained the same as previously reported, the authors selected erlotinib in lieu of lapatinib and fused it with the hydroxamic acid and benzamide motifs from the established HDAC inhibitors to develop the chimeric inhibitors [74,77].

Further investigation of the biological activity of the resulting hybrids revealed that the derivatives 11 and 12 in which erlotinib and the benzamide moiety were separated by the methylene linker resulted in the selective inhibition of HDAC class I isoforms including the successful inhibition of nuclear and cell HDAC (Table 12, Figure 16).

When tested against nuclear and cellular HDACs, compound **11** showed IC_50_ of 0.25 μM and 2.46 μM, respectively, and compound **12** behaved similarly with IC_50_ values of 0.2 and 1.85 μM, respectively (Table 12). It is to be noted that only these two benzamide derivatives exhibited dual inhibition of HDAC and EGFR along with other tumor cell lines with IC_50_ in the micromolar range (Table 13).

On the other hand, if the benzamide moiety was replaced with the hydroxamic acid in the inhibitors, the resulting derivative—compound **13** successfully inhibited both HDAC I and II isoforms in the nanomolar range along with nuclear and cellular HDACs with an IC_50_ of 6.4 and 3.4 nM, respectively (Table 12). Remarkably, these values were less than the reference compound SAHA, indicating the potency of the hybrid analog.

In addition to the dual inhibition of HDAC and EGFR, compound **13** was found to show cytotoxicity against all the tumor cell lines (HeLa, A549, A431, Cal27, SKBR3, and SKOV3) (Table 14). Although compounds **11** and **12** were also cytotoxic against other tumor cell lines, compound **13** was identified to be more potent from its IC_50_ values (Table 14).

In another attempt to circumvent the limitations in cancer treatment, Cai et al. synthesized a series of multi-acting HDAC/EGFR/HER2 inhibitors, and their research culminated in the development of CUDC-101—a potent, hybrid derivative of quinazoline based anti-cancer drug, combining HDAC inhibitory pharmacophore into EGFR and human epidermal growth factor receptor 2 (HER2) (Table 15, Figure 18) [78].

CUDC-101 has undergone Phase-I clinical trials for the treatment of solid tumors and head and neck squamous cell carcinoma, alongside tumor inhibition in various types of cancers such as breast, NSCLC, liver, head and neck, and pancreatic cancers [79,80,81].

The team studied the anti-proliferative effects of CUDC-101 on human acute promyelocytic leukemia (APL) and arsenic trioxide (ATO) resistant APL cell lines using a CCK-8 assay. The IC_50_ of CUDC-101 was found to be 0.6–5 µM, which indicated that NB4 and HL-60 cell lines were more sensitive to CUDC-101 than ATO. In addition, the drug evaluation on ATO-resistant APL cell lines revealed that, at a concentration of 0–5 µM, the drug was able to significantly inhibit the cell proliferation of NB4/As cells in a time-dependent manner [80]. All the above findings demonstrate that CUDC-101 is more potent over ATO in APL and ATO-resistant APL cell lines.

Following the above strategy, Zhang et al. designed and synthesized a series of dual inhibitors incorporating the pharmacophores of HDAC and EGFR inhibitors (Figure 2 and Figure 18) [82]. The so-called hybrids were designed based on the EGFR inhibitors: erlotinib and gefitinib and two established HDAC inhibitors: dacinostat and belinostat (Figure 2 and Figure 3). Furthermore, the activity of the resulting hybrids was tested against HDAC, EGFR, and HER2 receptors to evaluate the potency of the derivatives (Table 16 and Table 17). Studies revealed that the hydroxamate moiety is critical for potent HDAC activity and the linker aids in the zinc binding of the hydroxamate portion.

Although one of the hybrid compounds (compound **14**) showed anti-HDAC activity against HDAC 1, 2, and 6 with IC_50_ of 0.16, 0.18, and 0.56 µM, respectively, the same derivative was ineffective against EGFR and HER2 (Table 16) [82]. Overall, the SAR studies demonstrated that polar groups like hydroxamate (as in compound **14**) at 4th position of quinazoline ring will produce a potent HDAC/HER2 dual inhibitor rather than a HDAC/EGFR chimeric derivative.

Ding et al. used the similar concept to generate hybrid derivatives by combining the pharmacophores of the kinase inhibitors such as vandetanib, neratinib, BMS-690514, and TAK-285 with the HDAC inhibitor vorinostat (Figure 19) [83].

A triazole linker was used to append vorinostat to the kinase inhibitor and inhibition studies revealed that the resulting fusion hybrids exhibited different selectivity profiles relative to their parent compounds or individual inhibitors (Table 18 and Table 19).

Notably, one compound (**15**) demonstrated excellent inhibition of EGFR with IC_50_ of 10.3 nM and HDAC1 and HDAC6 with IC_50_ values of 1.1 and 4.3 nM, respectively (Table 19, Figure 20).

In addition to the inhibition of EGFR phosphorylation, compound **15** successfully demonstrated induction of hyperacetylation of histone H3 on the cellular level, and it showed excellent anti-proliferative activities against five cancer cell lines (A549, BT-474, A431, SK-BR-3, and NCI-H1975) with IC_50_ in the range of 0.2–7.8 µM (Table 20) [83].

Following a similar strategy, Goehringer et al. recently developed novel chimeric inhibitors targeting epidermal growth factor receptors (EGFRs) and HDACs [84]. These molecules were designed by coupling the quinazoline pharmacophore of EGFR inhibitors (e.g., gefitinib) and the zinc binding group of HDAC inhibitors like SAHA. Furthermore, the anti-cancer efficiency of the resulting hybrid derivatives was tested in prostrate and hepatocellular solid cancer cell lines in addition to leukemia/lymphoma cell models (Table 21) [84].

Of all the chimeric derivatives, compounds **16** and **17** demonstrated potent inhibition of tumor cell growth with IC_50_ values comparable to the reference compounds (**SAHA** and **gefitinib**) and, in some cases, even better than the individual or parent pharmacophores (Figure 20). These derivatives stand apart with their remarkable anti-proliferative, anti-angiogenic effects, and potent apoptotic induction in solid tumor cell models with less cytotoxic side-effects [84].

### 7.5. Quinazoline Based Dual Inhibition of HDAC and VEGFR

Vascular endothelial growth factor receptor (VEGFR) belongs to the family of tyrosine kinase receptors. VEGFR is involved in the regulation of both vasculogenesis and angiogenesis and is therefore not only essential for the physiological functions of normal cells but also plays a vital role in the pathophysiological conditions such as rheumatoid arthritis, inflammation, psoriasis, and in several types of cancers [85,86]. In the VEGFR family, VEGFR-2 and 3 are upregulated in most common human cancers and therefore form an attractive therapeutic target for cancer treatment. VEGFR-2 is activated by binding to VEGF-A which is essential for neovascular growth needed to support the tumor progression, while activation of VEGFR-3 induces lymphangiogenesis [87,88]. This binding triggers the signaling cascade that regulates cell survival, migration, differentiation into mature blood vessels and vasodialation.

Thus, drugs targeting VEGF signaling to inhibit angiogenesis have resulted in positive outcomes in certain cancers which include VEGF-A neutralizing antibody bevacizumab, ramucirumab, and small molecule inhibitors such as sorafenib, sunitinib, pazopanib, vandetanib, axitinib, regorafenib, cabozantinib, nintedanib, lenvatinib, and apatinib (Figure 21) [30,86,89,90]. FDA approval of these nine small molecule inhibitors demonstrates the anti-cancer efficacy of VEGFR-2 inhibitors.

Peng et al. designed and developed a series of dual hybrids by combining 4-anilinoquinazoline and hydroxamic acid targeting vascular endothelial growth factor (VEGFR) and HDAC (Figure 22) [91]. Although all the resulting hybrids were found to be potent inhibitors of HDAC and VEGFR, one compound in particular (compound **18**) was identified as most potent against VEGFR-2 and HDAC enzymes with nanomolar IC_50_ values (Table 22, Figure 23).

In addition, compound **18** was also found to be most potent against HDAC1, HDAC2, HDAC6, and HDAC8, and showed strong anti-proliferative activity against human breast cancer cell lines—MCF-7 lines. The experimental results were substantiated with the molecular docking studies, which demonstrated that compound **18** can be a potential HDAC and VEGFR inhibitor.

The same group developed dual inhibitors targeting VEGFR and HDAC, by integrating 4-amino-N-phenyl-quinazoline and hydroxamic acid moieties (Figure 22) [92]. Among the series, compound **19** turned out to be a best dual inhibitor with IC_50_ of 2.2 nM against HDAC and IC_50_ of 74 nM against VEGFR-2 (Table 23 and Table 24, Figure 23).

In addition, compound **19** showed excellent inhibition against breast cancer cell lines (MCF-7) with IC_50_ of 0.85 µM (Table 23). Molecular docking studies of the compound aligned well with the experimental results, and the results showed the compound interacted well with the active binding sites of HDLP and VEGFR-2, indicating the potential of compound **18** as an anti-cancer agent [92].

### 7.6. Quinazoline Based Dual Inhibitors of HDACs and Histone Methyltransferases (G9a)

Histone methylation is a complex post translational modification which involves the addition of one or more methyl groups from S-adenosylmethionine (SAM) to lysine and arginine residues on histone proteins [93]. This histone modification is a dynamic, irreversible process that is associated with chromatin structure, induction, or repression of gene expression and therefore is critical for cell development and differentiation, stress response, DNA damage repair and other important cellular events. Lysine methylation is a critical epigenetic modification regulating a number of cellular signal transduction pathways and is catalyzed by lysine methyltransferases and lysine demethylases [94]. G9a also known as lysine methyl transferase 1C or euchromatic histone lysine N-methyltransferase 2 (EHMT2) and G-9a like protein (GLP) also called as lysine methyl transferase 1D or euchromatic histone lysine N-methyltransferase (EHMT1) are critical epigenetic enzymes that catalyze the methyl transfer from SAM to the amino acid residues in histone and non-histone substrates [93,94,95,96]. Besides histone methylation, studies have shown that G9a and GLP are involved in the regulation of diverse cellular processes, DNA repair, chromatin remodeling and are overexpressed in a variety of cancers such as ovarian, lung, liver, breast, prostrate, leukemia, and colorectal cancers. Considering their unique physiological and pathophysiological functions, EHMT1/2 have emerged as cancer therapeutic targets and, as such, inhibitors targeting G9a and GLP evolved as a novel approach to cancer treatment [97,98,99]. Some of the examples of G9a/GLP inhibitors include BIX-01294, UNC0638, UNC0642, DS79932728, and compound **20** (Figure 24) [99,100,101].

Since both G9a/GLP and HDAC enzymes are overexpressed in several cancers and given their role in the tumor growth and progression, scientists developed dual inhibitors incorporating the pharmacophores of G9a/GLP and HDAC inhibitors to simultaneously block both the oncogenic targets to harvest the synergistic action of the resulting hybrids.

Zang et al. developed a series of dual inhibitors targeting HDAC and histone methyltransferases (G9a) [102]. After investigating the structural details of the HDAC and G9a inhibitors, the authors coupled the lipophilic quinazoline core with the zinc binding portion of the HDAC inhibitor using a variable linker (Figure 25).

The resulting hybrid molecules consisted of linker and hydroxamic acid at C2 and C4 position of the quinazoline ring.

In total, twenty different hybrid molecules were synthesized and cell-based assays were conducted to investigate their inhibition potential against several cell lines (Table 25).

Of all the synthesized compounds, two of them in particular, that include compounds **21** and **22**, showed potent inhibition against HDAC and G9a with activity comparable to BIX-01294 (Table 26, Figure 26).

Recently, Zheng et al. designed and synthesized a series of quinazoline based dual inhibitors targeting HDAC and GLP enzymes by integrating quinazoline pharmacophore with the zinc binding element of HDAC inhibitor (Figure 27) [103].

The resulting hybrid derivatives turned out to be very potent against GLP and HDAC1/6 enzymes with nanomolar IC_50_ values (IC_50_ < 190 nM) and cytotoxic against several cancer cell lines with micromolar IC_50_ values (IC_50_ < 50 µM) (Table 27 and Table 28).

Remarkably, among all the synthesized derivatives, compound **23** proved to be highly potent with the best dual inhibition ability demonstrated with an IC_50_ of 89 and 13 nM against HDAC1 and 6, respectively, and IC_50_ of 1.3 nM against GLP (Table 28, Figure 26).

In addition to the potent anti-proliferative effect against cancer cell lines, compound **23** was also capable of inhibition of HDAC and GLP on the protein level, G0/G1 arrest in HepG2cells, and induction of apoptosis in cancer cells.

## 8. Conclusions

Owing to the limitations and hurdles in the single drug therapy of cancer, the multi drug approach has evolved as an attractive tool for the drug discovery scientists to fight the battle against cancer. Given the tendency of cancer cell to bypass or outsmart any treatment, the technique of combination therapy is the most common method of treatment used in cancer chemotherapy. However, toxicity and undesired side effects from the use of multi-drugs is a major shortcoming with chemotherapy. In this context, the concept of molecular hybridization which involves the combination of two or more pharmacophores to obtain a hybrid drug has gained lot of attention in the past decade.

Despite the challenges involved in the design of the hybrid drugs like retaining the selectivity, pharmacological activity, and avoiding unfavorable drug–drug interactions, in most cases, hybrid drugs often proved to be more potent and showed high efficacy compared to the parent or individual pharmacophores. FDA approved quinazoline based drugs are reported to be successful in controlling the tumor growth and progression in several clinical studies. As the number of HDAC inhibitors gaining FDA approval for the treatment of various cancers has increased in the recent years, the combination of quinazoline and HDAC pharmacophores appears to be a promising approach to produce effective anti-cancer drugs. As suggested from different examples of dual HDAC inhibitors and their successful experimental results presented in this review, quinazoline based HDAC dual inhibitors show great potential to enter the clinical trials and soon emerge as effective anti-cancer tools.

## Figures and Tables

**Figure 1 molecules-27-02294-f001:**
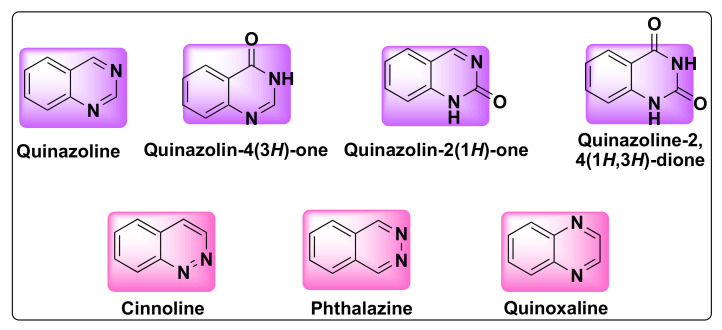
Structures of isomeric forms of quinazolines and its oxidized forms.

**Figure 2 molecules-27-02294-f002:**
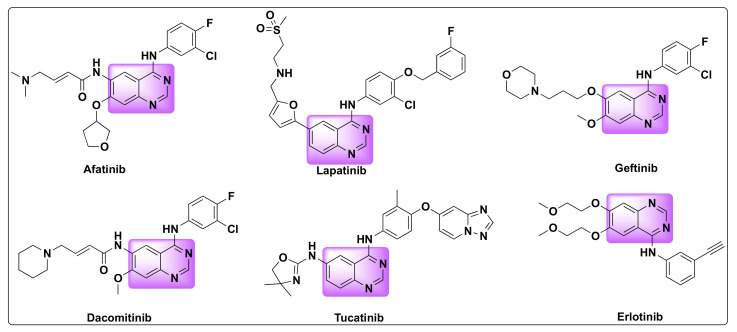
FDA approved quinazoline based drugs for the treatment of lung and breast cancer.

**Figure 3 molecules-27-02294-f003:**
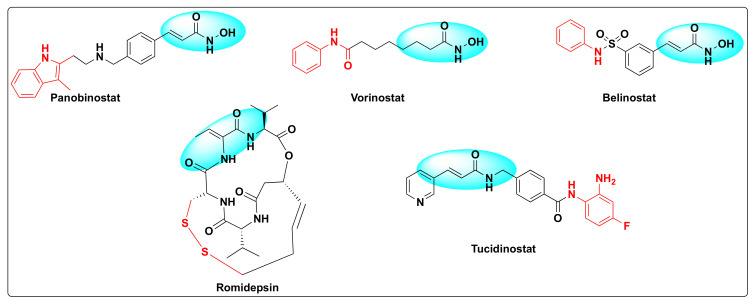
Some selected examples of HDAC inhibitors.

**Figure 4 molecules-27-02294-f004:**
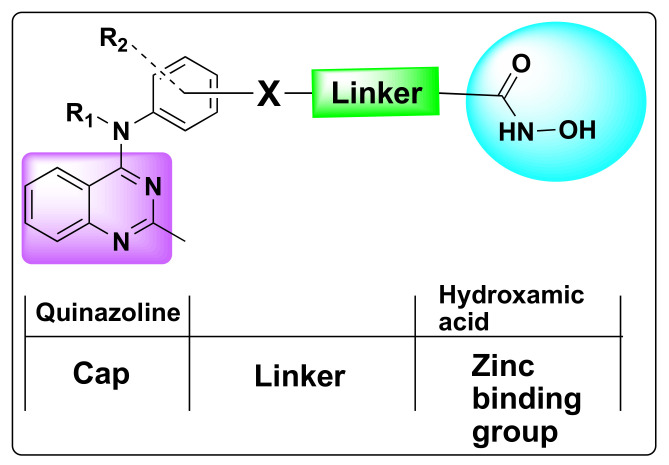
Design strategy of quinazoline based HDAC dual inhibitors.

**Figure 5 molecules-27-02294-f005:**
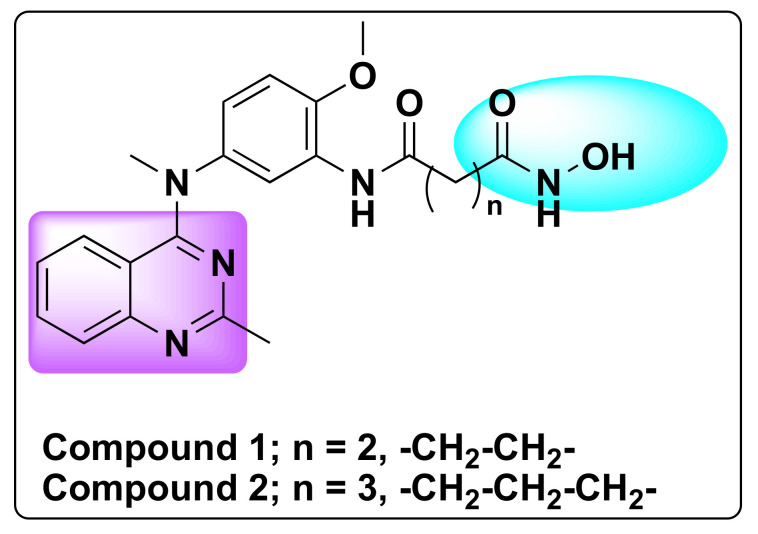
Chemical structure of compounds **1** and **2**.

**Figure 6 molecules-27-02294-f006:**
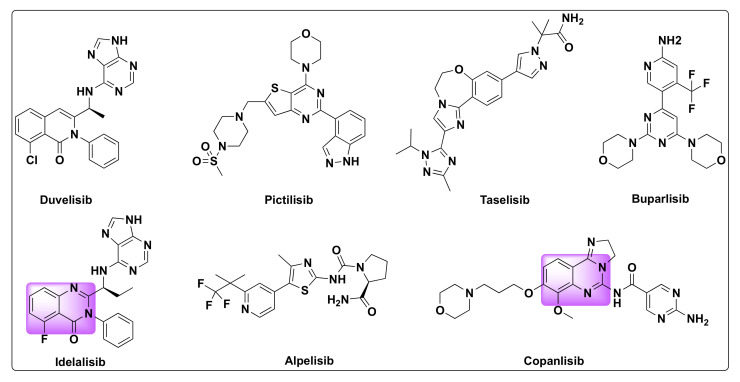
Some selected examples of PI3K inhibitors.

**Figure 7 molecules-27-02294-f007:**
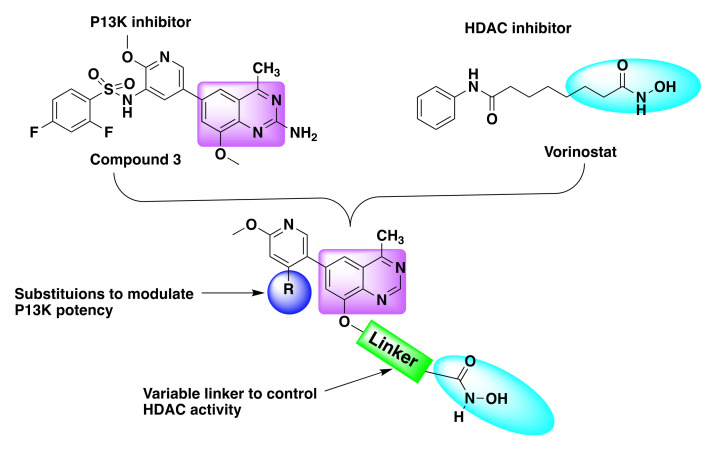
Design strategy of dual inhibitors.

**Figure 8 molecules-27-02294-f008:**
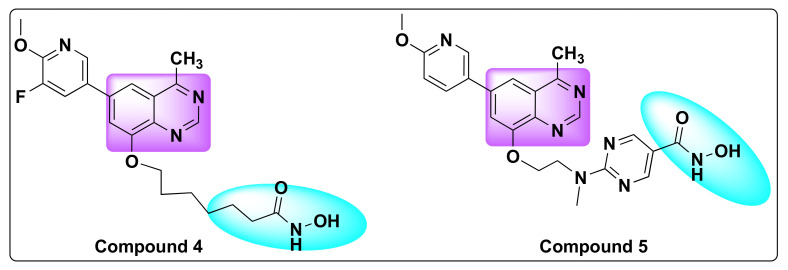
Chemical structures of compounds **4** and **5**.

**Figure 9 molecules-27-02294-f009:**
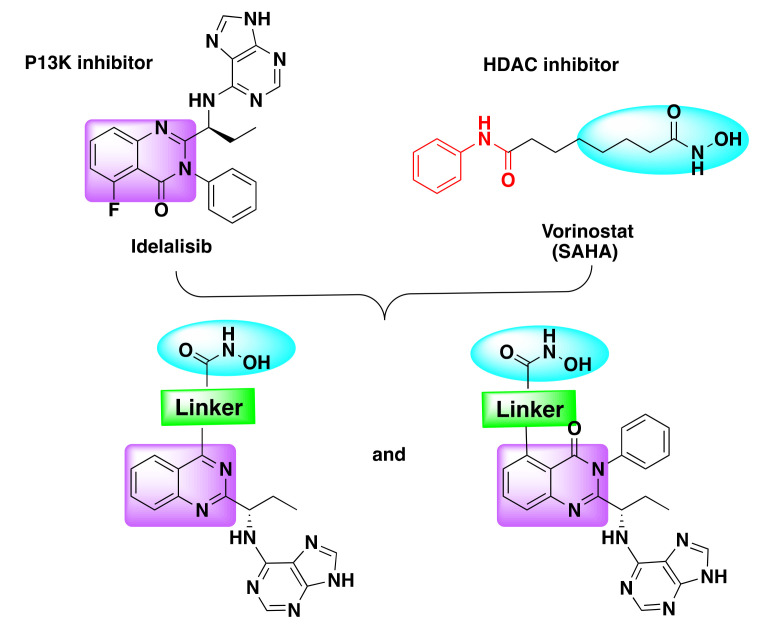
Design strategy of dual inhibitors.

**Figure 10 molecules-27-02294-f010:**
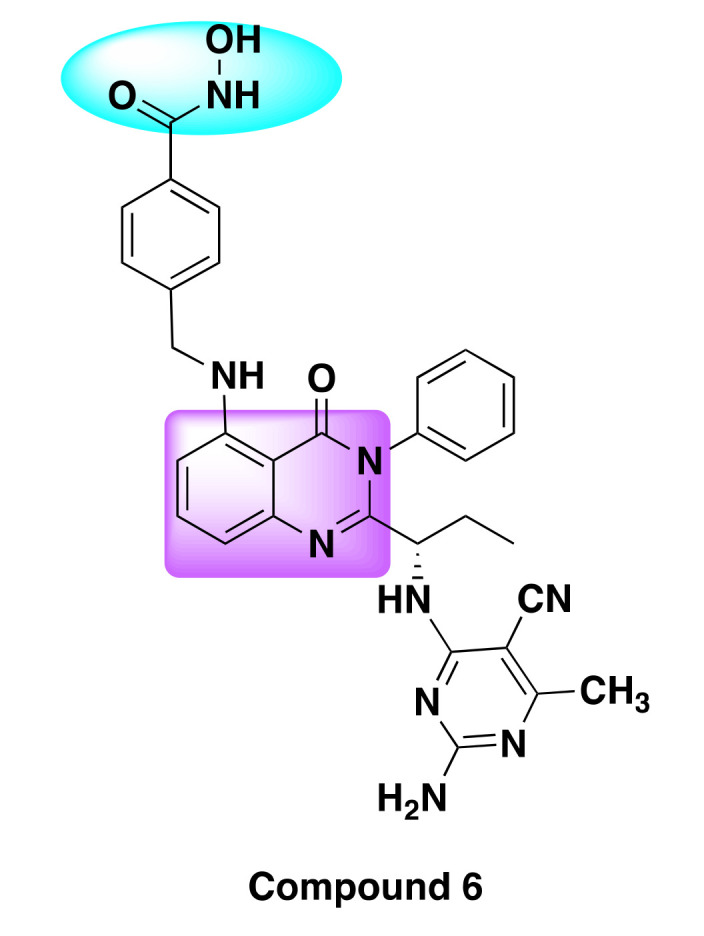
Chemical structure of compound **6**.

**Figure 11 molecules-27-02294-f011:**
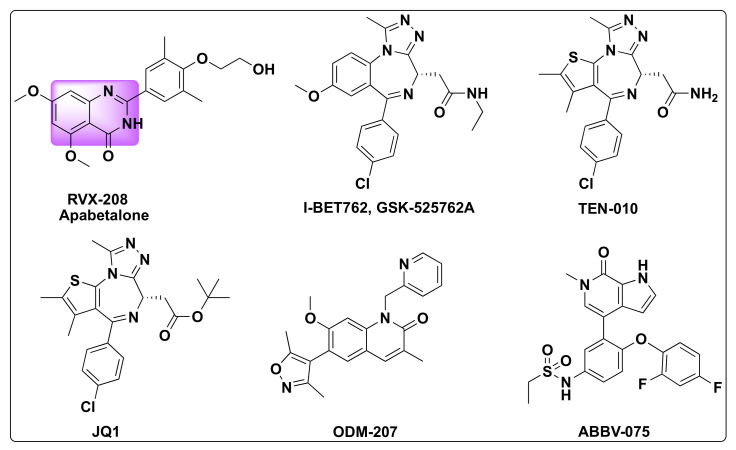
Some selected examples of BET inhibitors.

**Figure 12 molecules-27-02294-f012:**
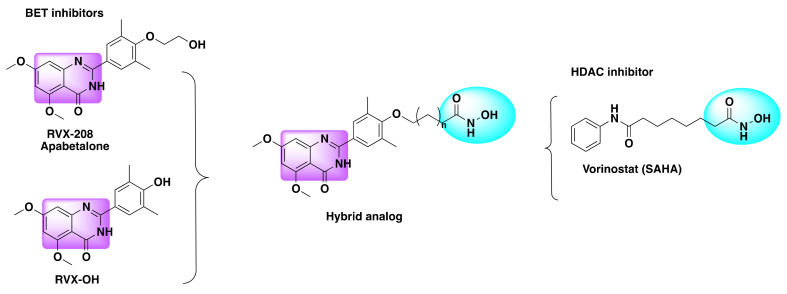
Design strategy of dual inhibitors.

**Figure 13 molecules-27-02294-f013:**
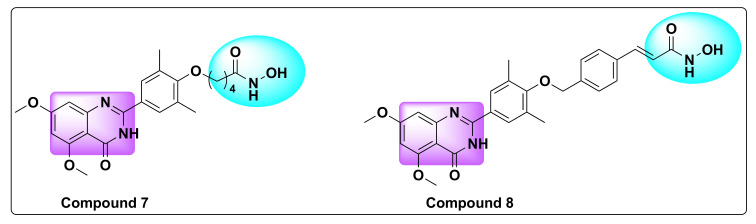
Chemical structure of compounds **7** and **8**.

**Figure 14 molecules-27-02294-f014:**
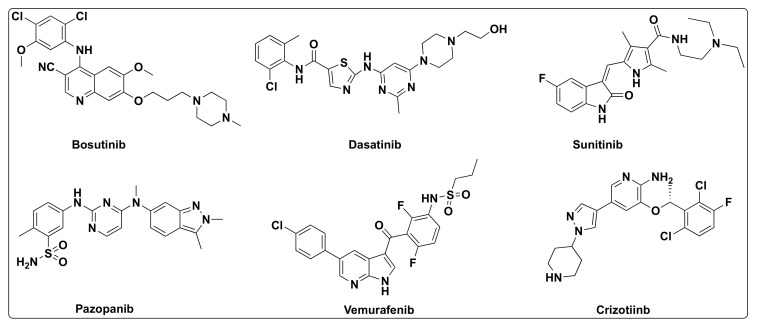
Chemical structures of EGFR inhibitors.

**Figure 15 molecules-27-02294-f015:**
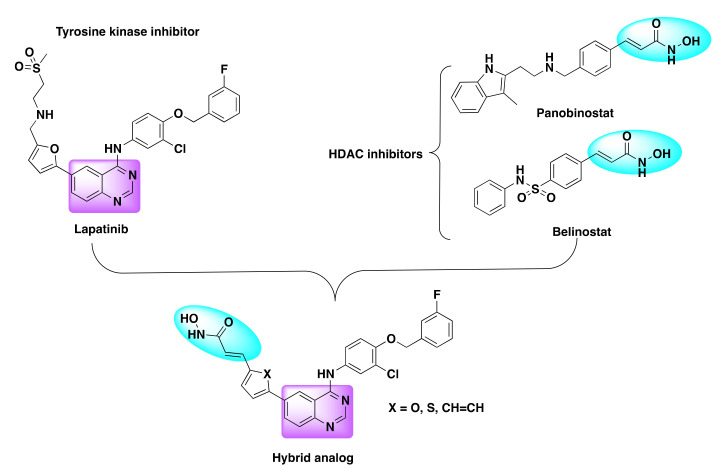
Design strategy of dual inhibitors.

**Figure 16 molecules-27-02294-f016:**
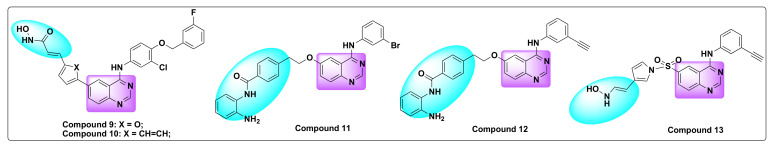
Chemical structures of compounds **9**–**13**.

**Figure 17 molecules-27-02294-f017:**
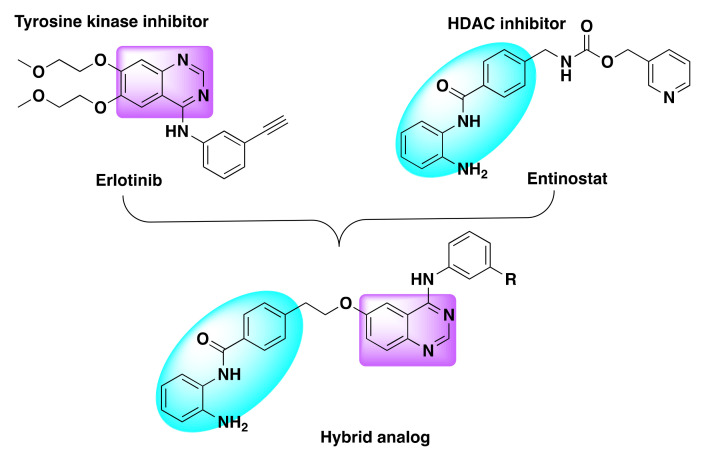
Design strategy of dual inhibitors.

**Figure 18 molecules-27-02294-f018:**
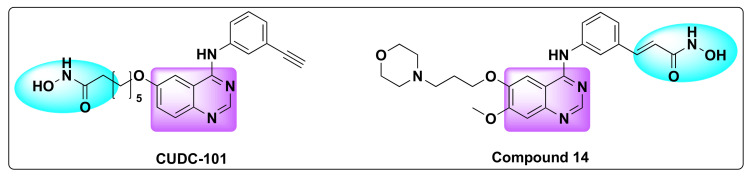
Chemical structures of CUDC-101 and compound **14**.

**Figure 19 molecules-27-02294-f019:**
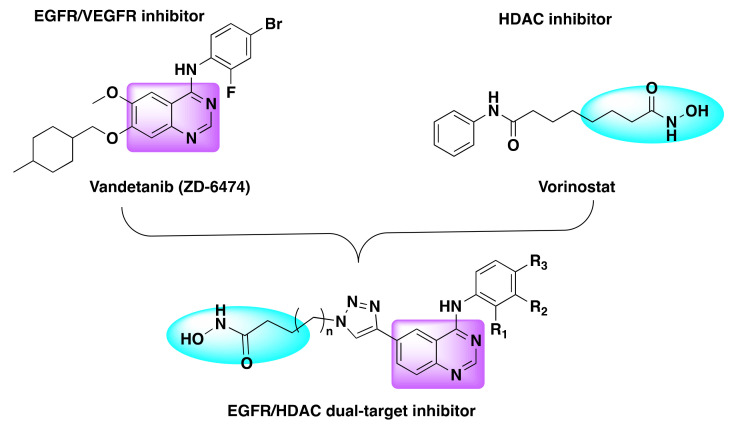
Design strategy of dual inhibitors.

**Figure 20 molecules-27-02294-f020:**
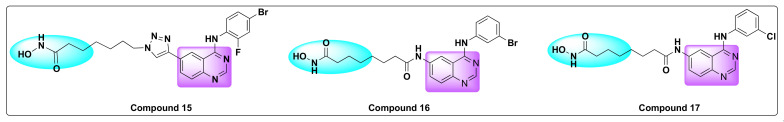
Chemical structures of compounds **15**–**17**.

**Figure 21 molecules-27-02294-f021:**
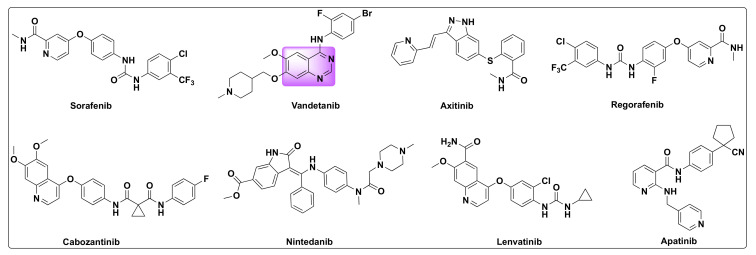
Some selected examples of VEGFR inhibitors.

**Figure 22 molecules-27-02294-f022:**
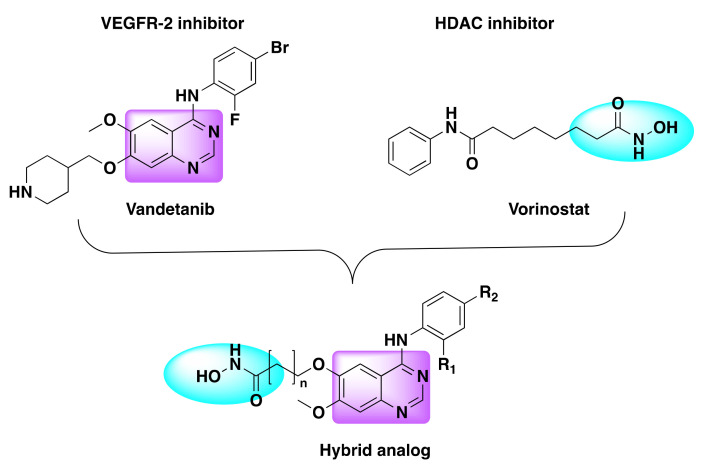
Design strategy of dual inhibitors.

**Figure 23 molecules-27-02294-f023:**
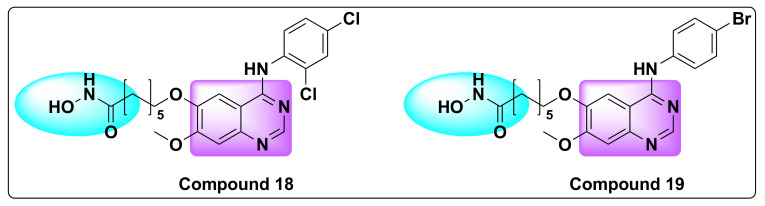
Chemical structures of compounds **18** and **19**.

**Figure 24 molecules-27-02294-f024:**
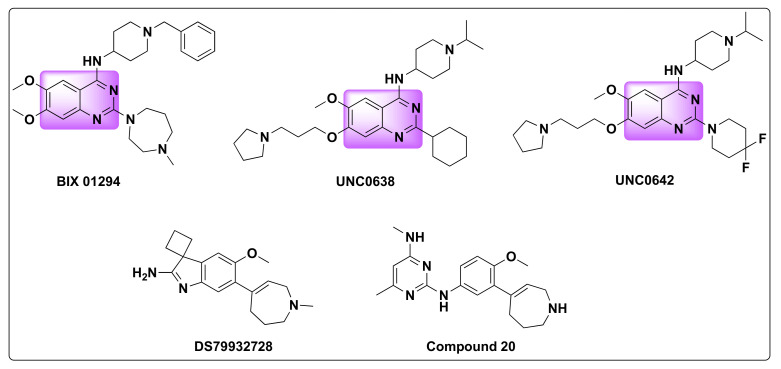
Some selected examples of G9a/GLP inhibitors.

**Figure 25 molecules-27-02294-f025:**
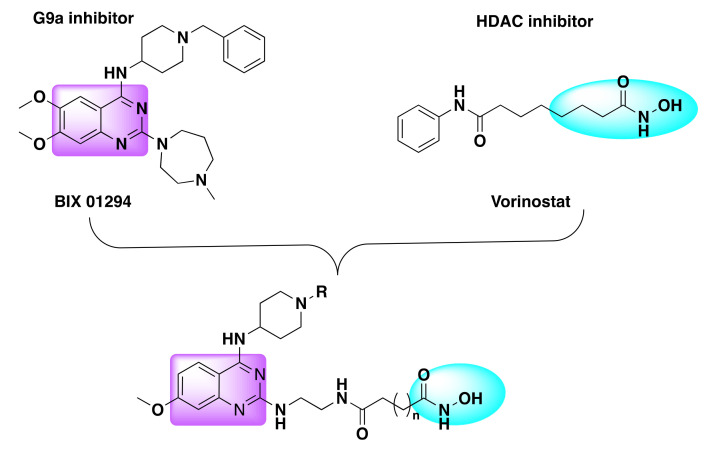
Design strategy of dual inhibitors.

**Figure 26 molecules-27-02294-f026:**
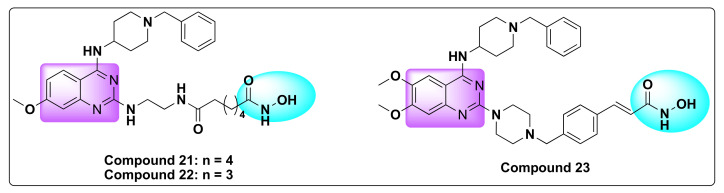
Chemical structures of compounds **21**–**23**.

**Figure 27 molecules-27-02294-f027:**
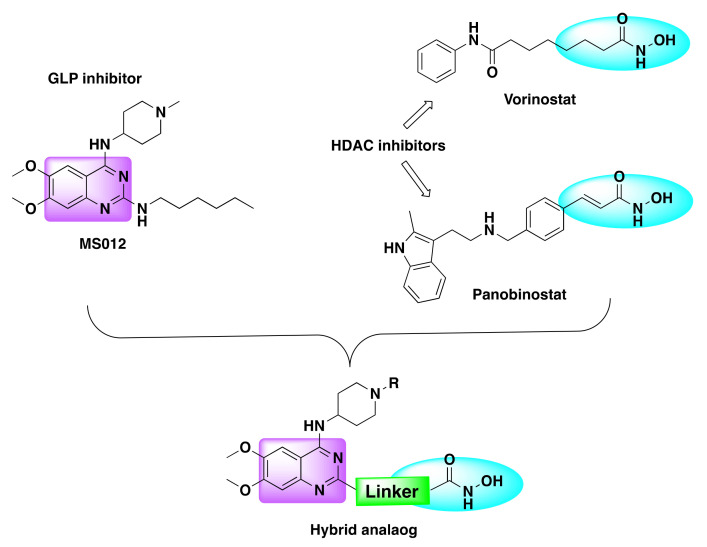
Design strategy of dual inhibitors.

**Table 1 molecules-27-02294-t001:** Inhibition of HDAC1 and HDAC6 by compounds **1**, **2** and **SAHA**.

Compd	IC_50_ (nM)	
	HDAC1	HDAC6
**1**	31 ± 0.37	16.15 ± 0.62
**2**	37 ± 0.24	35 ± 0.71
**SAHA**	11.4 ± 0.39	16.01 ± 1.37

**Table 2 molecules-27-02294-t002:** Anti-proliferative activity of compounds **1**, **2** and **SAHA** against various cell lines.

Compd	IC_50_ (nM)							
	HepG2	M-M-231	MCF-7	H1975	H460	Hela	U266	RPMI8226
**1**	3.5 ± 0.48	2.65 ± 0.71	2.55 ± 0.40	2.88 ± 0.69	1.05 ± 0.02	2.41 ± 0.59	<0.01	<0.01
**2**	4.51 ± 0.33	7.41 ± 0.81	8.62 ± 1.01	39.51 ± 1.00	37.59 ± 2.62	3.16 ± 0.15	0.15 ± 0.01	0.37 ± 0.21
**SAHA**	>1000	>1000	>1000	>1000	>1000	720 ± 13.02	569 ± 21.00	420 ± 9.37

**Table 3 molecules-27-02294-t003:** Inhibition of PI3Kα, HDAC1, mTOR and HCT116 by compounds **4**, **5**, **CUDC-907**, **BKM120**, and **SAHA** (Data undetermined is shown as “—”).

Compd	IC_50_ (nM)			IC_50_ (µM) HCT116
	PI3Kα	HDAC1	mTOR	
**4**	42	1.4	2861	0.15
**5**	226	1.1	6262	0.007
**CUDC-907**	69	0.36	431	0.005
**BKM120**	20	—	—	1.3
**SAHA**	—	32	—	1.9

**Table 4 molecules-27-02294-t004:** Anti-proliferative activity of compounds **4**, **5**, **CUDC-907**, **BKM120**, and **SAHA** against various cell lines (Data undetermined is shown as “—”).

Cancer Type	Cell Line	IC_50_ (µM)				
		Compd 4	Compd 5	SAHA	BKM120	CUDC-907
AML	THP-1	5.4	0.91	9.5	12	—
CML	K562	6.8	1.2	10	7.9	0.28
Breast cancer	MCF-7	2.0	1.1	4.1	5.7	0.041
	MDA-MB-453	0.30	0.004	0.37	0.37	0.009
Colon	HCT-8	0.32	0.059	0.35	1.7	0.005
	HCT116	0.15	0.007	1.9	1.3	0.005
Brain cancer	U87	3.2	0.34	10	4.8	0.007
NSCLC	NCI-H460	3.0	2.7	>100	3.2	0.15
Lung cancer	NCI-H1299	1.0	0.18	5.1	3.6	0.025
Pancreas cancer	Capan2	0.43	0.27	3.0	42	0.007
	SW1990	0.9	0.23	3.8	1.3	0.18
Prostrate	DU145	0.54	0.066	1.8	45	0.56
Stomach	HGC-27	0.11	0.014	0.35	0.99	0.041
Liver	HepG2	1.1	0.076	2.0	3.2	0.015
	Huh7	2.5	0.24	4.3	2.0	0.56
	BEL-7402	5.2	1.6	7.1	13	0.013

**Table 5 molecules-27-02294-t005:** Inhibition of PI3Kα, PI3Kβ, PI3Kγ, PI3Kδ, HDAC6, and HDAC isoform selectivity by compounds **6**, **PI-103**, and **TSA** (Data undetermined is shown as “—”).

Compd	IC_50_ (nM)					
	PI3Kα	PI3Kβ	PI3Kγ	PI3Kδ	HDAC6	HDAC Isoform Selectivity
**6**	47	—	9	7.0	12	>39
**PI-103**	4	7	55	5	—	—
**TSA**	—	—	—	—	6	—

**Table 6 molecules-27-02294-t006:** Anti-proliferative activity of compounds **6**, **Idelalisib**, and **SAHA** against various cancer cell lines.

Cancer Type	Cell Line	GI_50_ (µM)		
		Compd 6	Idelalisib	SAHA
Leukemia	CCRF-CEM	0.7	22.3	0.7
	SR	0.7		0.4
NSCLC	HOP-62	2.4	>100	1.6
	HOP-92	0.9	14.1	2.9
Colon cancer	HCT-2998	2.1	38.5	1.9
	KM-12	1.7	1.2	0.8
CNS cancer	SNB-75	0.7	1.2	0.8
	U251	1.7	53.2	1.6
Melanoma	LOX IMV1	1.6	33.5	1.2
	M14	1.8	37.8	1.3
Ovarian cancer	IGROV1	1.5	4.8	1.1
	OVCAR3	1.2	17.7	1.4
Renal cancer	A498	1.0	1.1	1.4
	CAKI-1	1.3	20.5	1.2
	RXF 393	1.2	1.4	1.3
Breast cancer	MDA-MB-232/ATCC	1.8	42.3	2.5
	HS 578T	2.0	6.0	3.6
	T-47D	1.2	5.2	0.5

**Table 7 molecules-27-02294-t007:** Inhibition of BRD4/BD1 and BRD4/BD2 by compounds **7**, **8**, **RVX-208**, and **SAHA TSA** (Data undetermined is shown as “—”).

Compd	IC_50_ (nM)		
	BRD4/BD1	BRD4/BD2	HDAC1
**7**	>5000	225 ± 32	32 ± 10
**8**	>5000	401 ± 21	204 ± 21
**RVX-208**	1985 ± 233	67 ± 9	—
**SAHA**	—	—	10 ± 3

**Table 8 molecules-27-02294-t008:** Activity of compounds **7** and **8** compared to standards **RVX-208** and **SAHA** against AML cell lines.

Compd	IC_50_ (µM)		
	MV4-11	OCI-AML2	OCI-AML3
**7**	1.67 ± 0.21	1.52 ± 0.21	1.09 ± 0.18
**8**	0.56 ± 0.09	0.38 ± 0.08	0.43 ± 0.18
**RVX-208**	4.48 ± 0.21	8.31 ± 0.32	7.17 ± 0.34
**SAHA**	0.98 ± 0.27	0.78 ± 0.32	0.85 ± 0.29

**Table 9 molecules-27-02294-t009:** Inhibition of HDAC HeLa nuclear extract, HDAC activity of recombinant HDAC isoenzymes, and induction of H3K hyperacylation by compounds **9** and **10**.

Compd	IC_50_ (µM)					
	HDAC	rHDAC1	rHDAC3	rHDAC6	rHDAC8	H3K (EC_50_)
**9**	0.63	0.61	6.30	0.23	11	3.65
**10**	0.047	0.035	0.066	0.086	0.63	4.70
**SNDX/MS 275**	11	0.082	0.62	>30	≥25	2.35

**Table 10 molecules-27-02294-t010:** Activity of compounds **9** and **10** in selected recombinant protein tyrosine and serine/threonine kinases.

Compd	IC_50_ (µM)							
	EGFR	HER2	PDGF-Rβ	InsR	Ab11	CDK2	PKA	PIK1
**9**	0.025	0.0041	3.3	7.34	7.7	0.44	120	12
**10**	0.018	0.011	2.5	3.6	5.3	0.92	>100	6.80
**Lapatinib**	0.0029	0.0045	8.5	17	23	11	>100	>100

**Table 11 molecules-27-02294-t011:** Anti-proliferative activity of compounds **9** and **10** against various cancer cell lines.

Compd	IC_50_ (µM)					
	A549	HeLa	A431	Cal27	SKOV3	SKBR3
**9**	3.9	3.2	1.9	0.82	2.1	0.51
**10**	1.2	0.97	0.72	0.59	1.0	0.35
**Lapatinib**	8.5	5.9	0.97	0.007	0.003	0.002
**SAHA**	1.8	1.5	4.2	3.2	2.2	2.6

**Table 12 molecules-27-02294-t012:** Inhibition of nuclear extract HDAC, cell HDAC and rHDAC1, rHDAC3, rHDAC6, and rHDAC8 enzymes by compounds **11**, **12**, **13**, **SAHA** and **erlotinib**.

Compd	IC_50_ (µM)					
	nHDAC	cHDAC	rHDAC1	rHDAC3	rHDAC6	rHDAC8
**11**	0.25	2.46	0.074	0.51	>32	>32
**12**	0.20	1.85	0.041	0.55	>32	>32
**13**	0.0064	0.034	0.0065	0.015	0.0073	0.59
**SAHA**	0.079	0.063	0.013	0.037	0.025	1.2
**Erlotinib**	>32	>15.8	>32	>32	>32	>32

**Table 13 molecules-27-02294-t013:** Inhibition of selected recombinant protein tyrosine and serine/threonine kinases in biochemical assays by compounds **11**, **12**, **13** and **erlotinib**.

Compd	IC_50_ (mM)							
	EGFR	HER2	PDGF-Rβ	InsR	Ab11	CDK2	PKA	PIK1
**11**	0.033	0.039	28.0	150	11.0	22.0	>100	>100
**12**	0.078	0.066	7.2	36	7.0	13.0	>100	>100
**13**	1.2	3.4	70.0	65.0	27.0	>100	>100	>100
**Erlotinib**	0.005	0.12	2.3	>100	2.1	>100	>100	>100

**Table 14 molecules-27-02294-t014:** Anti-proliferative activity of compounds **11**, **12**, **13**, **SAHA**, and **erlotinib** against various cancer cell lines.

Compd	IC_50_ (µM)					
	A549	HeLa	A431	Cal27	SKOV3	SKBR3
**11**	1.3	1.45	1.7	0.62	5.1	6.3
**12**	0.9	1.35	1.6	0.61	3.4	5.1
**13**	0.24	0.5	0.26	0.16	0.19	0.56
**SAHA**	1.8	1.5	4.2	3.2	2.6	2.2
**Erlotinib**	>50	8.3	2.5	0.074	15	31

**Table 15 molecules-27-02294-t015:** Activity of CUDC101, SAHA, erlotinib, and lapatinib against HDAC, EGFR, and HER2 receptors (Data undetermined is shown as “—”).

Compd	IC_50_ (nM)		
	HDAC	EGFR	HER2
**CUDC101**	4.4	2.4	15.7
**SAHA**	40.0	—	—
**Erlotinib**	—	48.0	134.8
**Lapatinib**	—	11.2	10.2

**Table 16 molecules-27-02294-t016:** Anti-proliferative activity of compound **14**, **SAHA**, and **lapatinib** (Data undetermined is shown as “—”).

Compd	IC_50_ (µM)		
	HDAC	HDAC3	HDAC6
**14**	0.16 ± 0.02	0.18 ± 0.05	0.56 ± 0.06
**SAHA**	0.25 ± 0.04	0.17 ± 0.02	0.23 ± 0.06
**Lapatinib**	—	—	—

**Table 17 molecules-27-02294-t017:** Comparison of RTK inhibition of compound **14** with lapatinib.

Compd	Invitro RTK Inhibition (%)	
	EGFR (%)	HER2 (%)
**14**	9.7	47.2
**SAHA**	0	0
**Lapatinib**	92.7	92.0

**Table 18 molecules-27-02294-t018:** Percent inhibition of VEGFR2, HER2 and EGFR by compound **15**, lapatinib, and staurosporine at 1 µmol/L (Data undetermined is shown as “—”).

Compd	VEGFR2 (%)	HER2 (%)	EGFR (%)
**15**	14.57 ± 26.24	−76.87 ± 4.49	99.26 ± 0.48
**Lapatinib**	—	91.38 ± 1.28	99.46 ± 1.14
**Staurosporine**	97.90 ± 1.78	—	—

**Table 19 molecules-27-02294-t019:** IC_50_ value of compound **15**, erlotinib, lapatinib, and SAHA against EGFR, HER2, HDAC1, and HDAC6 (Data undetermined is shown as “—”).

Compd	IC_50_ (nM)			
	EGFR	HER2	HDAC1	HDAC6
**15**	10.3 ± 0.8	>1000	1.1 ± 0.1	4.3 ± 0.2
**Erlotinib**	13.3 ± 0.9	—	—	—
**Lapatinib**	—	23.9 ± 1.4	—	
**SAHA**	—	—	12.2 ± 0.6	11.4 ± 1.1

**Table 20 molecules-27-02294-t020:** Anti-proliferative activity of compound **15**, **erlotinib**, and **SAHA** against A549, BT-474, A431, SK-BR-3, and NCI-H1975 cell lines.

Compd	IC_50_ (µM)				
	A549	BT-474	A431	SK-BR-3	NCI-H1975
**15**	0.71 ± 0.04	5.13 ± 0.62	0.26 ± 0.02	0.91 ± 0.05	7.85 ± 0.62
**Erlotinib**	16.83 ± 1.46	2.26 ± 0.17	1.85 ± 0.21	3.43 ± 0.19	23.76 ± 1.58
**SAHA**	2.57 ± 0.37	2.67 ± 0.38	2.29 ± 0.04	2.58 ± 0.13	1.90 ± 0.09

**Table 21 molecules-27-02294-t021:** IC_50_ values of compounds **16**, **17**, **SAHA**, and **gefitinib** in human prostate cancer (DU145), hepatoma (Hep-G2), and human T-cell lymphoma cell lines (Data undetermined is shown as “—”).

Compd	IC_50_ (µM)					
	DU145	Hep-G2	Jurkat	Hut78	SupT11	SMZ1
**16**	3.53 ± 0.23	4.94 ± 0.38	—	—	—	—
**17**	3.23 ± 0.18	3.92 ± 0.25	1.40 ± 0.12	1.18 ± 0.22	6.22 ± 0.25	2.24 ± 0.17
**SAHA**	0.68 ± 0.04	3.22 ± 0.44	1.7 ± 0.17	5.07 ± 0.42	4.67 ± 0.31	2.87 ± 0.33
**Gefitinib**	11.88 ± 2.13	18.53 ± 1.78	10.95 ± 0.28	>20	>20	>20

**Table 22 molecules-27-02294-t022:** Inhibitory activity of compound **18**, **vandetanib**, and **SAHA** against VEGFR2, HDAC enzymes, and MCF-7 (breast cancer) cell line.

Compd	IC_50_		
	VEGFR2 (nM)	HDAC (nM)	MCF-7 (µM)
**18**	84	2.8	1.2
**Vandetanib**	62	>10,000	18.5
**SAHA**	>10,000	12	4.5

**Table 23 molecules-27-02294-t023:** Inhibitory activity of compound **19**, **vandetanib**, and **SAHA** against VEGFR2, HDAC enzymes and MCF-7 (breast cancer) cell line.

Compd	IC_50_		
	VEGFR-2 (nM)	HDAC (nM)	MCF-7 (µM)
**19**	74	2.2	0.85
**Vandetanib**	54	>10,000	18.5
**SAHA**	>10,000	15	4.2

**Table 24 molecules-27-02294-t024:** Inhibition activity of compound **19** against HDAC isoenzymes (1, 2, 6, and 8).

Compd	IC_50_ (nM)			
	HDAC1	HDAC2	HDAC6	HDAC8
**19**	1.8	3.3	16.4	4.6

**Table 25 molecules-27-02294-t025:** Anti-proliferative activity of compounds **21** and **22** compared to standards BIX-01294 and SAHA against various cancer cell lines.

Compd	IC_50_ (µM)			
	MDA-MB-231	MCF-7	A549	HEK293
**21**	89.33 ± 1.23	79.43 ± 2.72	>100	56.96 ± 1.12
**22**	10.02 ± 1.66	37.36 ± 2.20	36.24 ± 1.76	19.95 ± 0.19
**BIX-01294**	2.155 ± 0.88	8.103 ± 1.99	21.74 ± 2.73	2.048 ± 0.98
**SAHA**	2.874 ± 0.84	8.124 ± 4.98	19.31 ± 1.26	2.482 ± 1.13

**Table 26 molecules-27-02294-t026:** IC_50_ data of compounds **21**, **22**, **BIX-01294**, and **SAHA** against G9a receptor, Hela, and K562 cells. (* G9a activity was assessed by H3K9Me2 cell immunofluorescence In-Cell Western (ICW) assay in (MDA-MB-231 cell line); ** Evaluation of HDAC activity in Hela and K562 using a homogeneous cellular assay) (Data undetermined is shown as “—”).

Compd	IC_50_ (µM)		
	G9a *	Hela **	K562 **
**21**	37.79 ± 2.80	15.33 ± 0.79	27.75 ± 0.59
**22**	7.136 ± 1.62	13.8 ± 1.22	5.735 ± 1.23
**BIX-01294**	4.563 ± 1.2	—	—
**SAHA**	—	5.044 ± 0.53	2.056 ± 0.59

**Table 27 molecules-27-02294-t027:** Anti-proliferative activity of compound **23**, **SAHA**, and **UNC0642** against various cancer cell lines.

Compd	IC_50_ (µM)							
	A431	HT-29	HeLa	K562	MDA-MB-231	HCT-116	HEPG2	HL-7702
**23**	2.38 ± 1.10	6.99 ± 2.77	3.34 ± 0.17	2.6 ± 0.95	9.3 ± 0.14	11.05 ± 0.04	2.74 ± 0.05	39.81 ± 8.86
**SAHA**	0.63 ± 2.03	0.56 ± 0.14	0.48 ± 0.57	0.68 ± 0.7	2.36 ± 0.36	1.08 ± 0.1	>50	1.35 ± 1.03
**UNC0642**	3.38 ± 1.22	1.35 ± 2.4	1.74 ± 0.07	0.92 ± 0.56	3.23 ± 0.28	5.01 ± 0.88	2.69 ± 0.06	3.23 ± 0.19

**Table 28 molecules-27-02294-t028:** Inhibition of HDAC enzymes and GLP by compound **23**, SAHA, and UNC0642 (Data undetermined is shown as “—”).

Compd	IC_50_ (nM)		
	HDAC1	HDAC6	GLP
**23**	89	13	1.3
**SAHA**	16	13	—
**UNC0642**	—	—	2.8

## Data Availability

Not applicable.

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
