# Peer review of "Quinazoline Based HDAC Dual Inhibitors as Potential Anti-Cancer Agents"

_molecules, 2022, doi:10.3390/molecules27072294_

Round 1
Reviewer 1 Report
- This review is acceptable. The authors provide a literature review of recent research articles to reiterate the quinazoline based HDAC dual inhibitors may be potential candidates of anti-cancer agents. It might be interesting and informative for readers of this field.
- In the draft, most figures were focused on the design strategies of inhibitors and their chemical structure. Diagramming of signaling pathways of how those quinazoline function as anti-cancer agents should be helpful for readers more quickly and easier to realize the topic of this review.
- Several typographic mistakes should be revised before submission.
Author Response
This review is acceptable. The authors provide a literature review of recent research articles to reiterate the quinazoline based HDAC dual inhibitors may be potential candidates of anti-cancer agents. It might be interesting and informative for readers of this field.
Authors’ response:
Thank you so much for your kind words of appreciation.
In the draft, most figures were focused on the design strategies of inhibitors and their chemical structure. Diagramming of signaling pathways of how those quinazoline function as anti-cancer agents should be helpful for readers more quickly and easier to realize the topic of this review.
Authors’ response:
Thank you for highlighting the need to include the signaling pathways. Considering this feedback and in response to another reviewers comment, we included tables (with biological activity) for all the classes of dual inhibitors discussed in the manuscript, that correlates the biological activity of the most efficient dual inhibitors to the reference anti-cancer agents with their target specific receptors/enzymes. In addition, their anti-proliferative activity against various cancer cell lines were also highlighted in separate tables. Unfortunately, with time constraints we could not draw/diagram the signaling pathways but we believe that these tables will be informative to the readers seeking information about the biological activity of these inhibitors with respect to the standard or approved anti-cancer agents.
Several typographic mistakes should be revised before submission.
Authors’ response:
Thank you very much for your comment. We have reviewed and fixed all the typographical errors in the manuscript.

Reviewer 2 Report
The manuscript titled "Quinazoline Based HDAC Dual Inhibitors As Potential Anti-Cancer Agents" is a very interesting review article describing the role of HDAC dual inhibitors. I think that the contents of the review are interesting, but some points need to be revised before publication:
-the title is not correct in my opinion. You describe the polypharmacological role of HDAC inhibitors with other targets, so you can change the title in an appropriate manner.
-Nevertheless, you can add a great part on the role of dual isoform HDAC inhibitors in the context of cancer.
-A table summarizing the more intriguing compounds and the cancer studied, must be useful to understand the potential of these compounds.
- a paragraph with clinical data must be useful for the multidisciplinary readership of the journal.
Author Response
The manuscript titled "Quinazoline Based HDAC Dual Inhibitors As Potential Anti-Cancer Agents" is a very interesting review article describing the role of HDAC dual inhibitors. I think that the contents of the review are interesting, but some points need to be revised before publication:
Authors’ response:
Thank you so much for your kind response and insightful thoughts on the manuscript.
-the title is not correct in my opinion. You describe the polypharmacological role of HDAC inhibitors with other targets, so you can change the title in an appropriate manner.
-Nevertheless, you can add a great part on the role of dual isoform HDAC inhibitors in the context of cancer.
-A table summarizing the more intriguing compounds and the cancer studied, must be useful to understand the potential of these compounds.
Authors’ response:
Thank you so much for your thoughtfulness and insights. We are very inspired with your comment to include tables for every example of dual inhibitor class discussed in the manuscript. These tables include the biological activity of the selected dual inhibitors with respect to the standard or approved HDAC inhibitor/anti-cancer agent. Also, anti-proliferative activity of these inhibitors against various cancer cell lines were presented in separate tables to inform the reader about the efficiency of these dual inhibitors as anti-cancer agents.
- a paragraph with clinical data must be useful for the multidisciplinary readership of the journal.
Clinical data????
Authors’ response:
Thank you for your comment. As described in the above comment, we hope that the new tables included in the manuscript can be informative to the readers seeking clinical information about the derivative.
